# Remote neuropsychological assessment of patients with neurological disorders and injuries—a study protocol for a cross-sectional case-control validation study

Nils Berginström ![ORCID],[1,2] Linus Andersson[2]

¹Department of Community Medicine and Rehabilitation, Rehabilitation Medicine, Umeå University, Umeå, Sweden
²Department of Psychology, Umeå University, Umeå, Sweden

**Correspondence to**
Dr Nils Berginström;
nils.berginstrom@umu.se

## ABSTRACT

**Introduction** There are great potential benefits of being able to conduct neuropsychological assessments remotely, especially for hard-to-reach or less mobile patient groups. Such tools need to be equivalent to standard tests done in the clinic and also easy to use in a variety of clinical populations.

**Methods and analysis** This study protocol describes a cross-sectional study aimed at validating the newly developed digitalized neuropsychological test battery Mindmore Remote in patients with neurological disorders and injuries. Diagnoses comprise traumatic brain injury, stroke, Parkinson's disease, multiple sclerosis, brain tumour and epilepsy. 50 patients in each patient group will be included. In addition, 50 healthy controls will be recruited. All participants will undergo both testing with Mindmore Remote at home and traditional neuropsychological assessment face-to-face in a randomised order. The primary outcome is the association between tests from the Mindmore Remote battery and their equivalent traditional neuropsychological tests. Further, bias between methods and differences between groups will also be investigated.

**Ethics and dissemination** The study protocol has been approved by the Swedish Ethical Review Authority (2022-06230-01) and adheres to the declaration of Helsinki. All participants will be given oral and written information about the study and sign informed consent forms before entering the study. All participants are informed that they can terminate their participation in the study at any given time, without giving any explanation, and participating in the study or not will not affect their care at the clinic. Neither authors nor personnel involved in the research project are affiliated with Mindmore AB. The results from the study will be published in peer-reviewed scientific journals and presented at national and international conferences on the topic.

**Trial registration number** NCT05819008.

## STRENGTHS AND LIMITATIONS OF THIS STUDY

⇒ A large cross-sectional study including 300 patients with different neurological disorders and injuries, and 50 healthy controls.
⇒ The patients are clinical patients currently being under evaluation for cognitive deficits, making the results from the study clinically generalizable.
⇒ If the remote test battery proves to be valid, such results would have direct practical implications in clinical practice.
⇒ Excluding participants with languages and cultural backgrounds other than Swedish limits the generalizability of the findings to the entire Swedish population.

situations where physical meetings are challenging. Remote healthcare solutions have always been a pertinent issue for patients from sparsely populated areas, who have a high degree of motor and/or cognitive disability, or who have difficulties or lack the resources to travel.

Patients with neurological conditions are a highlighted group in this regard. Due to injuries or progressive diseases within the central nervous system, they often struggle with logistical problems, either directly or indirectly related to their physical and mental symptoms. Diagnoses include traumatic brain injury,[1 2] stroke,[3] Parkinson's disease,[4] multiple sclerosis,[5] epilepsy[6] and brain tumours.[7] These conditions result in structural and functional disruptions to the brain and/or spine, causing both physical and cognitive disabilities. Deficits in cognitive and executive functioning are among the most common symptoms in patient groups. These characteristics render neuropsychological assessment a key part both in diagnosis and rehabilitation.[8]

Traditionally, the neuropsychological examination requires a physical meeting. The psychologist records the performance

## INTRODUCTION

Digital meetings are now a ubiquitous part of everyday life in many parts of the world, not least following the COVID-19 pandemic that necessitated widespread use of methods for online communication. These developments have the potential to benefit healthcare in

of the patient using paper-pencil test and a manual stopwatch. After the testing session, the psychologist manually scores the tests and compares the outcomes to normative performance using tables from testing manuals or peer-reviewed research articles. This is a time-consuming endeavour, with high costs for both patients and the healthcare system.

Computerised testing has become more common in the 21st century,[9 10] including combinations of paper-pencil tests and computers or tablets.[11 12] This has reduced the time that psychologists spend scoring results from neuropsychological tests. Traditional testing nevertheless requires the patient to physically visit a clinic for the assessment. During the last few years, there has been an increasing interest in performing neuropsychological assessment remotely.[13]

A core problem is that neuropsychological assessments are difficult to adapt to a digital, remote platform. Standardisation is a necessary component in many tests (see, eg, Delis *et al*[14] or Wechsler[15 16]) since results are compared with a normative population who have performed the tasks under the same conditions. Changing the administration procedure too much might make the results invalid. While some traditional tests such as simple verbal tasks may be administrated through, eg, video conference without major threats to validity, other tasks such as those requiring motor responses or visual tracking are ill-suited for such solutions.[17 18] There are some examples of neuropsychological testing platforms that also include automated administration and scoring procedures, such as Test My Brain Digital Neuropsychology Toolkit,[19] Philips IntelliSpace Cognition[20] or the Cogniciti's Brain Health Assessment.[21] It is, however, necessary to thoroughly validate such tools before they can be implemented in healthcare.

Mindmore (www.mindmore.com) is an automated digital application originally distributed to patients on-site at a clinic or healthcare provider using a tablet device. Its constituent tests have been validated in healthy adults[11] and include normative data scores.[22] A remote version (Mindmore Remote) is currently under development so that patients can complete neuropsychological testing procedures at home without any healthcare personnel present. Mindmore Remote is currently undergoing validation, and normative data are being collected from healthy adults (van den Hurk, personal communication, approved ethics application by the Swedish Ethical Review Authority, dnr 2019-02030). However, the instrument does currently not include data from patient groups with cognitive impairments. This protocol presents a research project aiming to validate Mindmore Remote in patients with neurological disorders and injuries.

## Objectives

The primary aim of this project is to assess the convergent validity of Mindmore Remote tests by comparing them with standardised neuropsychological tests. We pose the following research questions:

1. Are the tests in Mindmore Remote equivalent to traditional neuropsychological tests in patients with traumatic brain injury, stroke, multiple sclerosis, Parkinson's disease, epilepsy and brain tumour?
2. Can the results from Mindmore Remote be transferred into neuropsychological profiles (eg, the distribution of performance across several tests) that can be used for patients with Parkinson's disease and Multiple Sclerosis?
3. How do the patients experience undergoing a neuropsychological evaluation on their own compared with traditional neuropsychological assessment in a physical meeting with a psychologist?

## METHODS AND ANALYSIS

### Study design and setting

This study is a cross-sectional study with a case-control design. All reporting of results in peer-reviewed scientific journals will adhere to the STROBE guidelines.[23] The study is located at the Neuro-Head-Neck Centre (NHHC), including the departments of Neurology, Neurosurgery and Neurorehabilitation at Umeå University Hospital, Umeå. Data collection will be conducted during March 2023–December 2025.

### Participants

Patients with diagnoses of traumatic brain injury, stroke, Parkinson's disease, multiple sclerosis, epilepsy and glioma who are referred for neuropsychological assessment at NHHC will be offered to participate in the study. The aim is to include 50 patients in each diagnostic group.

Healthy adults (*n*=50) will be recruited as controls through advertising at Umeå University Hospital. Since the patient groups will differ in age variance, the aim is to recruit a control group with a variance in age. The controls will be reimbursed for their participation (100 SEK; approximately €10).

### Eligibility criteria

The inclusion and exclusion criteria are the same for patients and healthy controls, except that healthy controls should not be diagnosed with any neurological disorders.

#### Inclusion criteria
- Physician-generated neurological diagnosis of either traumatic brain injury, stroke, Parkinson's disease, multiple sclerosis, epilepsy and glioma.
- Age 18 or above
- In possession of a computer with internet connection.

#### Exclusion criteria
- Any other neurological diagnosis than the primary diagnosis (for healthy controls, any neurological diagnosis).
- Severe psychiatric disorders, including schizophrenia or other psychotic disorders, severe depression, bipolar disorder and suicidality.

**Table 1** Outcome measures

| Mindmore Remote test | Description | Main outcomes | Validity test |
|---|---|---|---|
| Symbol Digit Processing Test (SDPT) | SDPT is a test of attention and mental processing speed. The participant views a digit-symbol code key on the top of the screen. In the middle of the screen, one symbol appears. The task is to press the correct number as quickly as possible using the mouse on a number pad at the bottom of the screen. | Number of correct items coded over 90 seconds. | Coding from WAIS-IV (Wechsler, 2008) |
| Rey Auditory Verbal Learning Test (RAVLT) | RAVLT is a test of verbal learning and episodic memory. The participant will hear 15 words presented over five trials, with a recall task after each trial. After the learning trial, a distraction list of 15 words is administered, and after this, the task is to recall the first list once again. After 30 minutes (after Stroop below), a free recall retention trial is performed, and after this, a recognition trial, where the participant is supposed to identify the words of the first list in a list of 15 hits and 15 distractors. All responses are given verbally. | Total Learning—Number of words recalled during all five trials; Short-term Recall—Number of words recalled after the distraction list; Long-term recall—Number of words recalled after 30 minutes; Recognition—Number of correct hits and true negatives during recognition. | Word List Recall I and II from WMS-III[15] |
| Corsi Block | Corsi Block is a test of short-term and working memory. On the screen, 10 white squares will appear. These will blink in yellow in specific sequences of increasing length, starting with two items. Two different sequences will be presented of each length. After each sequence has finished, the participant is supposed to click with the mouse on the squares in the same order (Corsi Forwards) or in reverse order (Corsi Backwards). If the participant fails to produce the correct sequence in both trials on the same level, the test will end. Corsi Forwards is performed first and then Corsi Backwards. | Corsi Block Forwards—Number of correct trials in the forward condition; Corsi Block Backwards—Number of correct trials in the Backward condition. | Block Span from WMS-III[15] |
| Trail Making Test (TMT) – Click | TMT is a test of attention, visuomotor speed and the executive function of switching. The test consists of three conditions. In condition 1 (Number Sequencing), the participant will see numbers 1–25 in small circles on the screen and is supposed to click on the numbers in the correct order. In condition 2 (Switching), both numbers 1–13 and letters A–M appear in the circles. The participant is supposed to click on the letters and numbers in order, but each number and letter interchangeably – 1-A-2-B-3, etc. In condition 3 (Motor Speed), the circles are connected by a dotted line, and the participant is instructed to click on the circle in the same order as the line connects them. | Time to completion of each trial. | Trail Making Test from D-KEFS[14] |
| Stroop | Stroop is a test of the executive function inhibition, performed in two conditions. In condition 1 (Colour Naming and Reading), a colour word appears, printed in the same colour. The participant is instructed to click as quickly as possible on the correct colour word (blue, red, green or yellow), spelt out in black colour below the target word. In condition 2 (Inhibition), the colour words are written in a different colour. The participant is instructed to click, as quickly as possible, on the colour word written in black colour below the target word that matches the colour of the word, and not the word in itself. | Time to completion of each trial. | Color-Word Interference Test from D-KEFS[14] |
| FAS | In FAS, a test of verbal production and executive functions, the participant is instructed to say as many words as possible during 1 min, starting with a specific letter. The test is performed in three trials using letters F, A and S, respectively. | Number of words produced over all trials. | Verbal Fluency from D-KEFS[14] |

All tests will be performed using Swedish versions.
D-KEFS, Delis-Kaplan Executive Functions System; WAIS, Wechsler Adult Intelligence Scale; WMS, Wechsler Memory Scale.

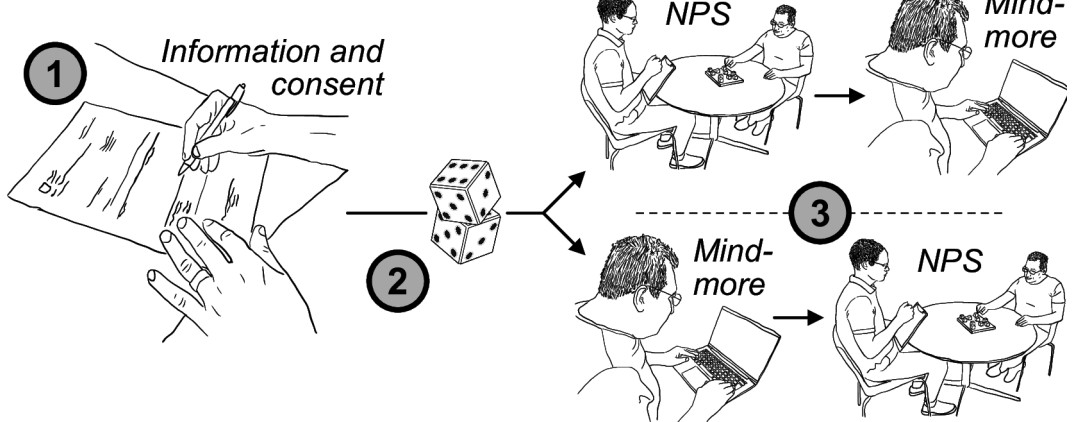

**Figure 1** Flow chart of the study. Note: After signing informed consent (1), participants will be randomised (2) into performing traditional neuropsychological examination (NPS) at the clinic first and then Mindmore Remote at home or vice versa (3).

► Addiction disorder, including both alcohol and other substances.
► Not speaking Swedish, have lived in Sweden less than 10 years and/or have had a majority of education outside of the Swedish educational system.
► Not being able to participate in a neuropsychological examination, or to give informed consent due to physical or mental disability.

### Instruments
#### Primary outcome measures: neuropsychological tests
All participants will be tested using all tests in the Mindmore Remote battery and corresponding traditional neuropsychological tests. Table 1 summarises all tests in Mindmore Remote and provides a detailed description of each test, primary outcome measures and the corresponding traditional neuropsychological test that will be used to test the validity of each Mindmore Remote test. The test battery will be performed in the order given in table 1, with the addition that the RAVLT Retention and Recognition trials will be performed after Stroop in the Mindmore Remote battery, and Word List Recall and Recognition Trials will be performed after Color-Word-Interference Test in the traditional battery.

Raw scores of the tests will be used as the main variables in the analyses. However, scaled scores (ie, those computed from the normative samples in the test manuals of the traditional tests and from the built-in norms in Mindmore Remote) might also be considered in some analyses.

#### Secondary outcome measures: self-assessment questionnaires
In addition to the neuropsychological tests, all participants will also complete paper-pencil versions of self-assessment questionnaires. The results from these questionnaires will mainly be used to adjust for the effects of possible confounders when analysing neuropsychological test data. The following self-assessment questionnaires will be used: Hospital Anxiety and Depression Scale (Zigmond and Snaith, 1983), Insomnia Severity Index (Bastien, Vallieres, & Morin, 2001), Cognitive Failure Questionnaire (Broadbent, Cooper, FitzGerald, & Parkes, 1982), Perceived Stress Scale (Levenstein et al., 1993) and Multidimensional Fatigue Inventory (Smets, Garssen, Bonke, & De Haes, 1995). Participants will also answer background questions regarding age, education, gender identity, occupation, time since injury/diagnosis, diagnosis, number of hours slept last night and medication, including medication during the last 24 hours.

### Data collection and management
#### Assessment procedure
After reading and signing informed consent forms, patients will be randomised into one of two test sequences – either beginning with performing the Mindmore Remote tests at home, followed by traditional neuropsychological testing on-site at the clinic, or vice versa (figure 1). Mindmore Remote testing will be performed by the participants in their homes after being contacted by the psychologist by phone. The remote session will not be directly observed by the psychologist. In cases where the data collection is part of a larger neuropsychological examination, the validation battery will be performed first during that examination to avoid the effects of fatigue. The test will be administered in the order outlined in table 1. Healthy controls will be randomised in a similar manner as patients, and except that data collection might be a part of a larger neuropsychological examination for patients, the data collection will not differ between groups.

All data collection will be performed by psychologists trained specifically for the task of neuropsychological examinations. The two test sessions will be performed within 2 weeks, with at least 1 day of resting in between. Each test session will last about 30–45 min, adding up to 1–1.5 hours of testing time for both assessments in total. As part of the traditional neuropsychological examination, patients will complete self-assessment questionnaires. After data collection, a subsample of 10 patients will be contacted and offered to participate in a semi-structured telephone interview regarding research question 3,

covering themes of usability and the difference of undergoing neuropsychological assessment on a computer at home versus at the clinic with a psychologist.

## Sample size

To be able to show that one patient group has a different cognitive profile compared with other patients or healthy controls, the aim is to include 50 participants in each group, which is the recommendation for sample size in normative studies of neuropsychological tests.[24] With alpha = 0.05 and a power of 80%, 16 participants would be enough to detect differences of one SD from the mean. With the goal of 50 participants in each group, this means a statistical power of 0.999 to detect differences of one SD from the mean, and a statistical power of 0.706 to detect differences of 0.5 SD from the mean.

## Data analysis plan

Validation of Mindmore test against traditional paper-pencil neuropsychological test will be performed using Pearson and/or Spearman correlation coefficients depending on the distribution of data. To evaluate bias between methods, the Bland-Altman method[25] will be used. In addition, multiple linear regressions will be performed to investigate and adjust for the influence of background variables, such as age, education, gender, diagnosis and time since injury/diagnosis. To establish neuropsychological profiles of diagnostic groups, descriptive data from each test will be presented, and analysis of variance, with and without covariates, will be performed to determine differences between groups. Due to multiple statistical analyses, adjustments for multiple comparisons will be considered.

## Patient and public involvement

Local patient organisations have been informed about the research project, and results from the study will be presented at meetings of these organisations.

## Ethics and dissemination

The study protocol has been approved by the Swedish Ethical Review Authority (2022-06230-01). Before initiating any form of neuropsychological assessment, all participants will be given oral and written information about the study and their participation and will sign informed consent. All participants are informed that they can abort their participation at any given time, without having to give reasons for this. For patients, participating in the study or not will not affect regular healthcare given at the clinic.

Undergoing a neuropsychological examination (remotely or traditionally) does not come with any risks in itself. However, it might be stressful and anxiety-provoking, especially for participants getting the impression that they are not performing well. This is a common clinical procedure, and the test leaders, all clinical psychologists, are used to handle such reactions during test sessions. In traditional testing, such fears can be identified and counteracted to a greater degree than during remote testing. We therefore collect reports of how a subsample experiences the testing situations pertaining to possible differences in negative affective states (objective 3).

## Data management

After inclusion, all participants will be assigned a code. All study material will be coded, and a code key will be stored separately in a secure compartment, only accessible to researchers within the project. Only personnel within the research project will have access to data. Study data will be stored for at least 10 years after completion of the study.

For Mindmore Remote, the data storage is sophisticated and secure, and no personal data are stored on Mindmore's servers. The invitation to the test is sent by a pseudonymized link, and only the researchers within the project have access to the code, as stated above. In this way, Mindmore AB cannot identify any person taking the test, although the test results are stored on their server.

## Plans for communicating important protocol amendments to relevant parties

In case the study protocol will undergo modifications or amendments, an additional ethical application will be submitted to the Swedish Ethical Review Authority. Accordingly, changes will be made in the trial registry ( Clinicaltrials.gov).

## Dissemination

Results from the study will be published in peer-reviewed open-access scientific journals. Furthermore, results will also be presented at national and international scientific conferences, as well as shared with patient organisations, healthcare officials, and policymakers.

**Acknowledgements** The authors would like to thank Malin Nygård, Sofia Näckter, Emma Wärn, Felix Sjögren and Therese Eriksson for assisting in data collection.

**Contributors** NB is the principal investigator and responsible for the project and for the reporting of results from the project. Data collection will be performed by clinical neuropsychologists at Neuro-Head-Neck Centre, Umeå University Hospital, all under the supervision of the principal investigator. NB wrote the initial draft of the manuscript. LA contributed to the study design and revised the manuscript for intellectual content.

**Funding** The study is supported by the Research and Development Fund granted by the County Council of Västerbotten [RV-978915], Stiftelsen Promobilia [A23004], the Swedish Neuropsychological Society [Grant number N/A] and a memorial gift to the County Council of Västerbotten regarding research at Umeå University Hospital [Grant number N/A].

**Competing interests** None declared.

**Patient and public involvement** Patients and/or the public were involved in the design, or conduct, or reporting, or dissemination plans of this research. Refer to the Methods section for further details.

**Patient consent for publication** Not applicable.

**Provenance and peer review** Not commissioned; externally peer reviewed.

**ORCID iD**
Nils Berginström http://orcid.org/0000-0003-1192-4527

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
