## [Reviewer comments · BMJ Open]

ARTICLE DETAILS

TITLE (PROVISIONAL)	Remote Neuropsychological Assessment of Patients With Neurological Disorders and Injuries – A study protocol for a cross-sectional case-control validation study
AUTHORS	Berginstrom, Nils; Andersson, Linus

VERSION 1 – REVIEW

REVIEWER	Eike Ines Wehling
REVIEW RETURNED	04-Feb-2024

GENERAL COMMENTS	BMJ Open-2023-080628 Remote Neuropsychological Assessment of Patients with Neurological Disorders and Injuries – A study protocol for a cross sectional case-control validation study Summary: The aim of the study is to validate an online digitalized neuropsychological test battery (Mindmore) by examining test results of the online battery with results from traditional well-established test in various patient groups. The study is registered in Clinical trials (April 2023) and data collection will be ongoing from March 2023-December 2025. General comments The objective of the study is important and should be highly welcomed. Online assessment is highly requested in particular in countries with a widespread population. Furthermore, this kind of assessment will unburden some neuropsychologists. The paper is well-written, I suggest some minor comments at the end but I have some comments to the authors which they may want to consider. - A neuropsychological assessment is barely only an assessment to receive a “value” but the process of assessment contains valuable information. In many cases the person assessed realizes shortcomings/impairment. During a session with either a neuropsychologist or technicians this is often an issue. In an online assessment, this aspect gets lost. Since the process of awareness is important for further treatment – what are or should ethical and methodological considerations be taken?- And how is the person tested taken care of after an assessment when he/she feels have performed badly? I see that this is addressed at the end – but maybe the authors could consider moving it into the introduction since a neuropsychological assessment is as said above more a process with immense value both for the NP and the person assessed.- The choice of participants and the data collected evokes some questions. To me, there are hardly “neuropsychological” profiles with regard to the patient groups listed. Furthermore, results from testing
---

	depend on time since injury/diagnosis, for some groups location of injury (glioma, stroke, TBI), severity of injury (NIHSS, GCS, tumor grad) etc. Further will there be differences regarding age and to some extend gender (verbal learning). What value have 50 patients from each group then to say something about the test?  - The authors indicate that “not speaking Swedish” as an exclusion criteria. How do they handle aphasia/language impairment that occurs in several of the diagnostic groups? How immigrants who have learned Swedish but may have a different cultural background. In what way have the authors considered a representative choice of the Swedish population? - Another issue not addressed is motor consequences. Several of the included patient groups will have motor impairments/reduced function. Have the authors any considerations about the differences between paper/pencil and mouse abilities? - What kind of validity intend the authors to analyze? What kind of criteria will be available after finishing the study? Normative data? Clinical cut off scores? - Due to the above named considerations, I am not sure if I can follow the estimation of sample size. A left-sided stroke patient will have a different profile than a right sided, same may apply TBI, a newly injured person will show a different profile than a person in a chronic state. - It would be ok to let the reader know some more statistics is that correlations? Intended cut-offs? Due to the multiple analyses – will there be adjustments considered? - The description of tests could use some adjustment for grammar and content. - Time – I assume the patients/participants will need at least 45 minutes Minor comments Some suggestions for rephrasing/wording and content  - Page 5 line 5 – who have a high degree of disability – please be more precises- do you think that individuals with severe cognitive impairment are included here? Disability due to mobility? - Delete “to neurons” line 9 - Why are cognitive and executive functioning named separately? Is this the classification suggested by Lezak et al., (2012)? - Page 6 line 49 – what validity? - Is the interview semi structured? What are the questions/topics that will be addressed?
--	---

REVIEWER	Ruth Sumpter NHS Greater Glasgow and Clyde
REVIEW RETURNED	15-Feb-2024

GENERAL COMMENTS	This is an important area of research and it is good to see a study proposal for publication. This is an area of professional interest to readers practising clinically in this field, and increased evidence base for this work is in demand. This is a real part of current clinical practice, particularly for hard to reach remote geography and disabled patients, and specific tools urgently require development and stringent evaluation in comparison to "traditional" face to face cognitive assessment tools. Minor changes/clarifications suggested:  • Throughout the document Mindmore is referred to as “Remote” or “Distance” – are these the same thing? • Can the authors declare any intellectual or financial interest in the development and or publication of Mindmore and profits around its
---

	future sale? It may be worth clarifying this under “Competing interests”?  • Can the authors make very clear whether Mindmore Remote is administered under observation of a clinician through videolink or completed independently via a URL by the patient independently? • Standardised approach to Mindmore Remote - How can authors be sure that patients will be able to follow instructions independently and ensure standardisation of testing environment (e.g. platform used, operating system used for timed tests, device used, kind of mouse used for responses – e.g. hand held mouse or laptop mouse, patient is the one who takes the test, no distractions in background, standard instructions for test environment etc)? My understanding is that timed tests for existing online/URL delivered self-assessment platforms (e.g. CANTAB) are problematic because of the varying devices, mouse and operating systems used causing varying timed responses. How will authors control for motor dysfunction on timed tests (e.g. delayed timed responses due to physical rather than cognitive speed)? • Page 3 - Line 25 - It is not clear whether "traditional neuropsychological assessment" will take place face to face or via remote administration. This is detailed and defined further on page 5, but it may be worth being clear in the abstract? • Page 5 Line 13 - Should read "cognitive disabilities" not "mental disabilities" • The methodology does not include comparison of Mindmore remote with Mindmore on site/in person testing, and I assume this has been addressed elsewhere - that is, has the remote version of Mindmore been already found to correlate with the remote version? • Page 9 Line 111 - It is not clear if these measures will be completed via pencil and paper or via a digital advice, or remotely on screen. It would be worth clarifying this, as digital administration/remote administration may not be validated? • Page 9 – Line 135 “In adjacent to the” would perhaps be better expressed as “As part of the” • Page 111 – Line 174 – typo “are us to” might be “are used to”? • There is at least one typographical error in Table 1 – may be worth proof reading.
--	---

VERSION 1 – AUTHOR RESPONSE

Reviewer: 1
Dr. Eike Ines Wehling

General comments

The objective of the study is important and should be highly welcomed. Online assessment is highly requested in particular in countries with a widespread population. Furthermore, this kind of assessment will unburden some neuropsychologists. The paper is well-written, I suggest some minor comments at the end but I have some comments to the authors which they may want to consider.
Author response: We thank the reviewer for the positive feedback on the objectives of the study.

- A neuropsychological assessment is barely only an assessment to receive a “value” but the process of assessment contains valuable information. In many cases the person assessed realizes shortcomings/impairment. During a session with either a neuropsychologist or technicians this is often an issue. In an online assessment, this aspect gets lost. Since the process of awareness is important for further treatment – what are or should ethical and methodological considerations be taken?
- And how is the person tested taken care of after an assessment when he/she feels have performed badly? I see that this is addressed at the end – but maybe the authors could consider moving it into

the introduction since a neuropsychological assessment is as said above more a process with immense value both for the NP and the person assessed.

Author response: You describe a core challenge regarding remote neuropsychological assessment, which will affect not only this research project, but the entire field of remote neuropsychological assessment, both in research and clinical practice. We do believe that the issues you raise must be addressed for remote testing to be feasible, but also humbly argue that is not within the scope of the research aim of this study, but rather an important ethical issue. We therefore hope that you find it reasonable that we keep the issue in the ethics section.

Hence, we argue that the process and qualitative aspect of the neuropsychological assessment is beyond the scope of this particular aim, despite it being of fundamental importance. Here, we study another fundamental issue that has to do with the quantitative properties of the tests in Mindmore Remote and their traditional counterparts. Thus, this is not something that will be discussed at length in this manuscript, but we agree with the reviewer that this is an important aspect of remote testing that needs to be addressed. We therefore clarify how we collect and use telephone interview data regarding this issue. (p 11, line 182-185) "In the traditional testing, such fears can be identified and counteracted to a greater degree than during remote testing. We therefore collect reports of how a sub-sample experience the testing situations pertaining to possible differences in negative affective states (objective 3)"

- The choice of participants and the data collected evokes some questions. To me, there are hardly "neuropsychological" profiles with regard to the patient groups listed. Furthermore, results from testing depend on time since injury/diagnosis, for some groups location of injury (glioma, stroke, TBI), severity of injury (NIHSS, GCS, tumor grad) etc. Further will there be differences regarding age and to some extent gender (verbal learning). What value have 50 patients from each group then to say something about the test?

Author response: We agree with you that for some patient groups, presenting neuropsychological profile may not be suitable. However, especially for patients with Parkinson's Disease and Multiple Sclerosis, a neuropsychological profile in Mindmore Remote might be useful for the clinician. Research question 2 has been changed accordingly (p 6, line 56-57) "2. Can the results from Mindmore Distance Remote be transferred into neuropsychological profiles (e.g., the distribution of performance across several tests) that can be used for patients with Parkinsons' Disease and Multiple Sclerosis?"

- The authors indicate that "not speaking Swedish" as an exclusion criteria. How do they handle aphasia/language impairment that occurs in several of the diagnostic groups? How immigrants who have learned Swedish but may have a different cultural background. In what way have the authors considered a representative choice of the Swedish population?

Author response: The practical exclusion criteria are "not speaking Swedish" and "have lived in Sweden less than 10 years" and/or "have had a majority of education outside of the Swedish educational system". The exclusion criteria in the manuscript have been changed accordingly (P 7, line 92-93).

You raise an important concern regarding cultural background, which is a problem for the entire neuropsychological field. However, including cultural/language aspects is beyond the scope of the current aim (but may be an important issue for future studies). As you point out, this will affect generalizability to the entire Swedish population as it is constituted today, and, we have included this as a limitation in the Strength and limitation section (p 3, "Excluding participants with languages and cultural backgrounds other than Swedish limits the generalizability of the findings to the entire Swedish population."

- Another issue not addressed is motor consequences. Several of the included patient groups will have motor impairments/reduced function. Have the authors any considerations about the differences between paper/pencil and mouse abilities?

Author response: Motor dysfunction vs cognitive dysfunction will be tested in the Mindmore Remote TMT-Click test, which also includes a motor task (condition 3), as stated in table 1. This task is similar to the motor task in TMT from D-KEFS (condition 5). Both these tests will be performed within the project and analyzed accordingly. However, there are no motor control task for other tests.

- What kind of validity intend the authors to analyze? What kind of criteria will be available after finishing the study? Normative data? Clinical cut off scores?

Author response: The study is primarily a convergent validity study, which is now stated in the aims section (p 5, line 49). In the data analysis plan we have included a section on which data will be available (p 10, line 163): “descriptive data from each test will be presented, and”. However, since neuropsychological assessments more often rely on interpretation of performance using normative data rather than cut-offs, suggested cut-offs will not be presented.

- Due to the above named considerations, I am not sure if I can follow the estimation of sample size. A left-sided stroke patient will have a different profile than a right sided, same may apply TBI, a newly injured person will show a different profile than a person in a chronic state.

Author response: As stated above, the power calculation is based on the possibility of detecting differences between groups (different patient groups and healthy controls) in test that are supposed to measure cognitive functions commonly affected after neurological injuries/disorders. Data on diagnosis and time since injury/diagnosis will be collected, and possibly used as covariates in the analyses. This is now also stated in the data analysis plan (p 10, line 162): “In addition, multiple linear regressions will be performed to investigate and adjust for the influence of background variables, such as age, education, gender, diagnosis, and time since injury/diagnosis.”

- It would be ok to let the reader know some more statistics is that correlations? Intended cut-offs? Due to the multiple analyses – will there be adjustments considered?

Author response: We thank the reviewer for this comment. Adjustments for multiple comparisons will be considered, and a statement about this have been added to the text: (p 10, line 165-166) “Due to multiple statistical analyses, adjustments for multiple comparisons will be considered.” We also added a line on presentation of descriptive data from each test (p 10, line 163) “descriptive data from each test will be presented, and”. However, since neuropsychological assessments more often rely on interpretation of performance using normative data rather than cut-offs, suggested cut-offs will not be presented.

- The description of tests could use some adjustment for grammar and content.

Author response: We thank the reviewer for noticing these errors. Table 1 has now been proofread and changed accordingly.

- Time – I assume the patients/participants will need at least 45 minutes

Author response: This is an accurate estimate regarding the traditional assessment, but the remote session last about 30 minutes: The text has been changed accordingly: (p 9, line 139-140) “Each test session will last about 30-45 minutes, adding up to 1-1,5 hours of testing time for both assessments in total.”

Minor comments

Some suggestions for rephrasing/wording and content

- Page 5 line 5 – who have a high degree of disability – please be more precises- do you think that individuals with severe cognitive impairment are included here? Disability due to mobility?

Author response: We thank the reviewer for noticing this unclarity in the manuscript. We have now stated that this includes both motor and cognitive disability (p 4, line 5-7): “Remote healthcare solutions have always been a pertinent issue for patients from sparsely populated areas, who have a

high degree of motor and/or cognitive disability, or who have difficulties or lack the resources to travel.”

- Delete “to neurons” line 9

Author response: These words have now been deleted.

- Why are cognitive and executive functioning named separately? Is this the classification suggested by Lezak et al., (2012)?

Author response: Yes, Lezak et al. (2012) actually state that executive functions “differ from cognitive functions in a number of ways” (p 37), and thus should be named separately.

- Page 6 line 49 – what validity?

Author response: We thank the reviewer for noticing this omission. We have now included “convergent” to clarify that the study assesses primarily convergent validity (p 5, line 49).

- Is the interview semi structured? What are the questions/topics that will be addressed?

Author response: We thank the reviewer for pointing this out. A clarification have been made (p 9, line 141-144): “After data collection, a subsample of 10 patients will be contacted and offered to participate in a semi-structured telephone interview regarding research question 3, covering themes of usability and the difference of undergoing neuropsychological assessment on a computer at home vs at the clinic with a psychologist.”

Reviewer: 2

Dr. Ruth Sumpter, NHS Greater Glasgow and Clyde

Comments to the Author:

This is an important area of research and it is good to see a study proposal for publication. This is an area of professional interest to readers practising clinically in this field, and increased evidence base for this work is in demand. This is a real part of current clinical practice, particularly for hard to reach remote geography and disabled patients, and specific tools urgently require development and stringent evaluation in comparison to “traditional” face to face cognitive assessment tools.

Author response: We thank you for the positive feedback on the objectives of the study.

Minor changes/clarifications suggested:

- Throughout the document Mindmore is referred to as “Remote” or “Distance” – are these the same thing?

Author response: Thank you for noticing this inconsistency. This has now been changed, and we are using Mindmore Remote throughout the manuscript.

- Can the authors declare any intellectual or financial interest in the development and or publication of Mindmore and profits around its future sale? It may be worth clarifying this under “Competing interests”?

Author response: We are thankful for you pointing this out. A clarification has been made under Competing interests which now states (p 12, line 218-219): “None declared. Neither authors nor personnel involved in the research project are affiliated with Mindmore AB.”

- Can the authors make very clear whether Mindmore Remote is administered under observation of a clinician through videolink or completed independently via a URL by the patient independently?

Author response: We agree this should be clarified. Participants will not be observed during testing, and under assessment procedure the text now states (p 9, line 129-131): “Mindmore Remote testing

will be performed by the participants in their homes after being contacted by the psychologist by phone. The remote session will not be directly observed by the psychologist.”

- Standardised approach to Mindmore Remote - How can authors be sure that patients will be able to follow instructions independently and ensure standardisation of testing environment (e.g. platform used, operating system used for timed tests, device used, kind of mouse used for responses – e.g. hand held mouse or laptop mouse, patient is the one who takes the test, no distractions in background, standard instructions for test environment etc)? My understanding is that timed tests for existing online/URL delivered self-assessment platforms (e.g. CANTAB) are problematic because of the varying devices, mouse and operating systems used causing varying timed responses. How will authors control for motor dysfunction on timed tests (e.g. delayed timed responses due to physical rather than cognitive speed)?

Author response: This is what we actually try to investigate through this study: Is Mindmore Remote a feasible tool for assessing cognitive functioning in patients without another person being present? Participants will use different computers, input devices, operating systems etc. It should also be noted that in traditional testing, the error of margin is quite large due to the fact that time is measured using a manual stop watch. If we fail to find relationships between the test in Mindmore Remote and the traditional tests, the conclusion might be that using Mindmore Remote in the way which it is intended to might not work. The project does adhere to the recommendations for digital neuropsychological testing from the Swedish Neuropsychological Association.

Motor dysfunction vs cognitive dysfunction will be tested in the TMT-Click test, which also includes a motor task (condition 3), as stated in table 1. This task is similar to the motor task in TMT from D-KEFS (condition 5). Both these tests will be performed within the project. However, there are no motor control task for other tests.

- Page 3 - Line 25 - It is not clear whether "traditional neuropsychological assessment" will take place face to face or via remote administration. This is detailed and defined further on page 5, but it may be worth being clear in the abstract?

Author response: We agree that this should be clear in the abstract, and have now included this: “All participants will undergo both testing with Mindmore Remote at home and traditional neuropsychological assessment face-to-face.”

- Page 5 Line 13 - Should read "cognitive disabilities" not "mental disabilities"

Author response: We thank you for noticing this error, and have now changed the text accordingly.

- The methodology does not include comparison of Mindmore remote with Mindmore on site/in person testing, and I assume this has been addressed elsewhere - that is, has the remote version of Mindmore been already found to correlate with the remote version?

Author response: You raise an important issue, and these comparisons have been made by Mindmore themselves, however, the results have not been published in peer-reviewed articles and are only available in the manual of the test. The aim of the current project is more encompassing, by investigating Mindmore Remote with traditional “gold standard” neuropsychological tests, which we believe will better indicate if the tests in Mindmore Remote are valid neuropsychological measures.

- Page 9 Line 111 - It is not clear if these measures will be completed via pencil and paper or via a digital advice, or remotely on screen. It would be worth clarifying this, as digital administration/remote administration may not be validated?

Author response: Thank you for pointing this out, this has now been clarified (p 8, line 114): “participants will also complete paper-pencil versions of self-assessment questionnaires.”

- Page 9 – Line 135 “In adjacent to the” would perhaps be better expressed as “As part of the”

Author response: We thank you for this suggestion and have now changed the text accordingly.

• Page 111 – Line 174 – typo “are us to” might be “are used to”?
Author response: Thanks for noticing this typo. We have now changed it accordingly.

• There is at least one typographical error in Table 1 – may be worth proof reading.
Author response: Thanks! Table 1 has now been proofread and changed accordingly.

VERSION 2 – REVIEW

REVIEWER	Eike Ines Wehling
REVIEW RETURNED	09-Apr-2024

GENERAL COMMENTS	Thanks for good replies to the stated comments and congratulations to a valuable project.
---